# HTK-N: Modified Histidine-Tryptophan-Ketoglutarate Solution—A Promising New Tool in Solid Organ Preservation

**DOI:** 10.3390/ijms21186468

**Published:** 2020-09-04

**Authors:** Annika Mohr, Jens G. Brockmann, Felix Becker

**Affiliations:** Department of General, Visceral and Transplant Surgery, University Hospital Muenster, 48149 Muenster, Germany; Jens.Brockmann@ukmuenster.de (J.G.B.); Felix.Becker@ukmuenster.de (F.B.)

**Keywords:** organ transplantation, organ preservation, ischemia reperfusion injury, histidine-tryptophan-ketoglutarate, HTK-N

## Abstract

To ameliorate ischemia-induced graft injury, optimal organ preservation remains a critical hallmark event in solid organ transplantation. Although numerous preservation solutions are in use, they still have functional limitations. Here, we present a concise review of a modified Histidine-Tryptophan-Ketoglutarate (HTK) solution, named HTK-N. Its composition differs from standard HTK solution, carrying larger antioxidative capacity and providing inherent toxicity as well as improved tolerance to cold aiming to attenuate cold storage injury in organ transplantation. The amino acids glycine, alanine and arginine were supplemented, N-acetyl-histidine partially replaced histidine, and aspartate and lactobionate substituted chloride. Several in vitro studies confirmed the superiority of HTK-N in comparison to HTK, being tested in vivo in animal models for liver, kidney, pancreas, small bowel, heart and lung transplantation to adjust ingredients for required conditions, as well as to determine its innocuousness, applicability and potential advantages. HTK-N solution has proven to be advantageous especially in the preservation of liver and heart grafts in vivo and in vitro. Thus, ongoing clinical trials and further studies in large animal models and consequently in humans are inevitable to show its ability minimizing ischemia-induced graft injury in the sequel of organ transplantation.

## 1. Introduction

Organ shortage remains the Achilles heel of solid organ transplantation. There is an eminent unmet need to develop strategies to overcome the continuously growing imbalance between demand and supply in transplant medicine, which could be attenuated by increasing the donor pool or prolonging the lifetime of transplanted organs. Among the various strategies aiming to remedy these problems are structural improvements (e.g., increasing the general awareness and willingness towards organ donation), logistical refinements (e.g., enhance timing of transplantation) as well as approaches of donor reconditioning, enhancement of post-transplant care and especially the use of machine perfusion [1,2,3,4,5].

However, the single most complex pathophysiological process during organ transplantation remains the inevitable ischemia reperfusion injury (IRI). At present a period of ischemia is unavoidable in solid organ transplantation and, therefore, organs are axiomatically subjected to IRI [6]. IRI is an interdependent two-phased process consisting of a first period of ischemia-induced cellular damage based on oxygen and nutrient deprivation during organ storage; followed by a second injury phase during reperfusion (inflammatory response, ROS production, calcium overload, correction of pH, mitochondrial dysfunction) [7,8,9]. IRI is of superior relevance in transplant medicine, as it significantly affects both the available donor pool as well as the lifespan of transplanted organs. IRI affects the current donor pool since especially marginal organs are known to suffer severely from IRI, adding additional risk of graft failure [10,11,12]. In order to utilize more marginal organs from extended criteria donors, further improvements are necessary to ameliorate IRI. Moreover, while there is sufficient data linking IRI to short-term graft function (e.g., delayed graft function in kidney transplantation), mounting evidence also suggests a strong influence on long term graft fade [13,14,15]. Thus, the beginning programs the end, which makes IRI and organ preservation an ideal target to overcome the hallmark problem of transplant medicine, organ shortage.

While the field of organ preservation is rapidly evolving with ongoing research and emerging clinical trials supporting a new concept of dynamic organ preservation, one of the key elements in organ preservation remains to seek for the optimal preservation solution.

With the rise of solid organ transplantation in the 1970s, a myriad of preservation solutions have been developed including the Euro-Collins (EC), Celsior (CE), University of Wisconsin (UW) and Histidine-Tryptophan-Ketoglutarate (HTK) solution. Although their composition differs (Table 1), their preservation principles are analogous. First, these solutions counteract graft edema due to sodium and water retention during ischemia and hypothermia with substances containing osmotic properties (e.g., glucose, mannitol, lactobionate, raffinose) [16]. Secondly, the addition of antioxidants (e.g., allopurinol, glutathione, tryptophan, mannitol, histidine) reduce the burden of oxygen free radicals. Thirdly, by buffering (e.g., phosphate, bicarbonate, histidine) metabolic acidosis occurring from anaerobic metabolism and adenosine 5′triphosphate (ATP) hydrolysis is counteracted. Fourthly, addition of energy precursors such as adenosine or glutamic acid/glutamate induce higher levels of ATP and improved mitochondrial function [17,18].

Several studies have aimed to determine optimal preservation solution for different organs with respect to graft and patient survival [19,20,21,22,23]. Nowadays, standard solutions for solid organ preservation are UW and HTK [24,25,26]. Despite equivocal results regarding their efficacy in a number of studies, the above-mentioned solutions still maintain functional limitations. In 2008 Rauen et al. assessed the inherent effects of six different preservation solutions in cultured hepatocytes. Especially, HTK showed marked toxicity observed during cold incubation, with histidine being the most likely responsible agent [27]. To ameliorate protective effects of the preservation solution, the authors suggested replacing injurious components when designing future preservation solutions.

Based on these findings, a novel preservation solution, termed HTK-N, has been developed by modifying the original HTK-solution [28,29]. Its composition differs from the former solution regarding its antioxidative capacity, inherent toxicity as well as superior cell protection aiming to further attenuate cold storage injury in organ transplantation. This review will first give a comprehensive view onto the modifications of HTK-N and its possible impact on preserved organs. Subsequently, existing studies evaluating HTK-N solution in solid organ transplantation will be summarized.

## 2. Composition of HTK-N Solution

Several potentially improving changes have been made to the original HTK-solution when developing the modified version. In general, glycine, alanine and arginine were supplemented. In addition, N-acetyl-histidine partially replaced histidine, and aspartate and lactobionate substituted chloride (Table 2, Figure 1).

In the beginning of its development, Rauen et al., were conducting experiments to assess mechanisms of histidine-induced cell injury and to provide alternative options for the retention of high buffering capacities [28]. They showed that histidine toxicity is mediated by an iron-dependent pathway. By entering cells, histidine forms redox-active iron complexes with intracellular iron inducing generation of reactive oxygen species (ROS). Based on these results, membrane-permeable iron chelators (e.g., deferoxamine, LK614) were supplemented to improve the preservation solution [30]. In addition, histidine derivative N-acetyl-histidine was found to exhibit comparable buffering capacities to histidine under culture conditions but less toxicity, probably due to its decreased cellular uptake. Therefore, histidine was partially substituted by N-acetyl-histidine [28,29,31].

The first modifications of the original HTK solution also involved the addition of glycine and alanine. Glycine showed significant protection under in vitro hypoxic conditions for cells of different origins and for isolated organs ex vivo as well as in animal studies [32,33,34,35]. These beneficial effects might be due to the prevention of hypoxia-induced sodium influx by inhibiting opening of chloride channels or by impeding formation of non-specific pores for small ions. Similar results have been reported for alanine [33,35].

Aspartate was added to potentially produce ATP. By supplementing both aspartate and alpha-ketoglutarate (already present in HTK solution) citric acid cycle metabolites are provided [29,31,32]. Thus, production of ATP by anaerobic glycolysis could prevent generalized cell injury during ischemia.

Lactobionate, a substrate known from UW solution, was supposed to provide a protective effect due to its ability to suppress cell swelling during cold storage [36] and was, therefore, supplemented as a novel substrate to the HTK-N solution.

In addition, arginine has been recognized to have favorable attributes to be added to the HTK-N solution. Microcirculatory changes are common and occur during reperfusion [37]. Arginine, as a substrate for nitric oxide synthase, could counteract these by contributing to vasodilatation secondary to the production of nitric oxide [38]. As this amino acid is commonly depleted during cold storage, its supplementation could attenuate microcirculatory ischemic injury [39].

To evaluate superiority of HTK-N solution, it was tested in different organ systems of animal models in vivo to adjust the ingredients of HTK-N solution to the required organ specific conditions, as well as to determine its innocuousness, applicability and advantages (overview see Table 3).

### 2.1. Thoracic Organs

#### 2.1.1. Heart

Originally developed for cardioplegia, HTK was found to be superior in myocardial preservation for heart transplantation when compared to other cardioplegic solutions [43,56]. Nevertheless, preservation of donor hearts after cardioplegic arrest is still limited to 3–6 h in contrast to most abdominal organs [44]. To overcome myocardial and endothelial damage during static cold storage (SCS), several studies were conducted to evaluate HTK-N and its optimal composition in heart transplantation models. Cold storage and the amino acid histidine has been found to increase intracellular levels of redox-active chelatable iron [57,58,59]. Further cold-induced, iron-independent injuries are mediated by chloride [27], thus, first modifications in experiments with HTK consisted of a chloride poor HTK variant in which histidine was replaced with N-acetyl-l-histidine and l-arginin was supplemented [40]. In a heterotopic heart transplant model three different compositions of HTK were tested: (a) addition of l-arginine and N-a-acetyl-l-histidine; (b) addition of iron chelators deferoxamine and LK-614, (c) traditional HTK as control. After 1 h of ischemic preservation, the hearts were implanted intraabdominally into the recipient rat and myocardial function and energy charge potential was analyzed after 1 h of reperfusion. The modified HTK solution with the addition of l-arginine and N-acteyl-l-histidine significantly improved myocardial contractility and relaxation after heart transplantation. Interestingly, the addition of iron chelators in the second tested solution diminished the beneficial effects, which the authors assigned to the relatively high concentration.

Next, a chloride-poor compared to a chloride-rich variant of the novel HTK solution was evaluated in isogeneic cardiac mouse grafts to potentially further improve its components [43]. Similar to the former study of Koch et al., histidine was partially replaced by N-acetyl-histidine. Furthermore, glycine, alanine (inhibition of sodium influx), arginine (NO synthase substrate), aspartate (additional energy substrate) and iron chelators (deferoxamine and LK 614) were added to the novel HTK-solution. After a cold storage of 24 h and a reperfusion and survival time of 2.5 months, the chloride-rich variant showed a significantly longer graft survival and attenuated myocardial injury, thus, suggesting the presence of chloride ions to be crucial for heart preservation [43]. Moreover, the study could show that the usage of iron chelators improved parameters of cell function (e.g., enzyme release of creatine kinase (CK) and lactate dehydrogenase (LDH)) and decreased parameters of cell injury. In addition, iron chelators decreased the generation of cardiac F2-isoprostanes, a highly specific and sensitive biomarker of oxidative stress. Nevertheless, as mentioned before it has to be noted that iron chelators in high concentrations have a negative effect on heart preservation due to toxicity [41,43].

When evaluating the final composed HTK-N in heterotopic heart transplantation of rats and a reperfusion time of 1 or 24 h, a significantly better result with HTK-N could be demonstrated especially in the early phase of reperfusion (1 h). Left ventricular systolic pressure and coronary blood flow was significantly higher, whereas the number of apoptotic cells was clearly reduced in the HTK-N group. The authors concluded that the better preserved endothelial function might be partly accountable for the fast myocardial recovery [42].

For prolonged ischemia times (15 h, 24 h) in a mouse cardiac transplant model the novel HTK-N solution resulted in attenuated cold-induced myocardial injury demonstrated by decreased histological evidence of injury, less interstitial fibrosis and less vasculopathy [43,44]. In addition to that, re-beating time and palpation score were significantly ameliorated [44] and grafts stored in HTK-N showed improved graft survival in contrast to the traditional HTK solution [43].

To better understand the protective effect of HTK-N three different ischemia conditions were investigated in isolated guinea pig hearts: 80 min ischemia at 30 °C, 81 min ischemia at 30 °C with short intermittent perfusion of the cold cardioplegic solution and 360 min ischemia at 5 °C [45]. The authors could demonstrate best results for HTK-N after warm ischemia (80 and 81 min) with shortest reestablishment of cell coupling, best heart rhythm and highest left ventricular developed pressure probably due to the high calcium level in HTK-N solution. In contrast, HTK showed superior results for long-lasting ischemia under hypothermic conditions.

The obtained results in rodent models and the optimal composition of HTK-N for heart transplantation now needs to be confirmed in non-rodent animal models and in humans to demonstrate the superiority of HTK-N.

#### 2.1.2. Lung

A recent in vitro study (using cultured lung epithelial cells) provided strong evidence for an ameliorated cold-induced injury and improved cell protection when using HTK-N [60]. Based on these results, Pizanis et al., tested HTK-N in a porcine model of single-lung transplantation with prolonged cold storage times. Porcine lung grafts were flushed with either the current clinical gold standard low-potassium dextran solution (LPD, Perfadex), HTK-N, HTK-N without iron chelators or HTK-N supplemented with dextran 40 [46]. After 24 h cold storage in the respective solution, left lungs were transplanted into recipient animals, the pulmonary artery and bronchus were clamped to solely evaluated the graft function and animals were kept alive for 6 h. While the authors could demonstrate equal effects of HTK-N compared to the gold standard LPD solution, addition of dextran 40 led to a significant improvement of post-transplant lung function. This was evident in a reduction of mean pulmonary arterial pressure and pulmonary vascular resistance as well as a decline of oxidative stress. Since the lung is stored in the presence of oxygen, oxidative stress in the form of ROS formation is a relevant problem during cold storage, and thus the presence of iron chelators in HTK-N may provide a potential explanation for the protective mechanism. Moreover, this is an elegant study since it provides first evidence for an organ tailored approach in the development of organ perfusion solutions. By adding dextran 40 (which is already established in clinical lung transplantation) to HTK-N significant improvements were reported. While HTK-N alone was not superior to HTK or LPD, this study suggests that HTK-N can be used a basic solution to which further organ specific supplements can be added.

### 2.2. Abdominal Organs

#### 2.2.1. Liver

First modifications in the composition of HTK were tested in hepatocytes under culture conditions [28,30]. As the impact of histidine resulted in loss of viability within 3 h especially under hypoxic conditions, Rauen et al. tested antioxidants such as N-acteylcysteine as well as iron chelators and histidine derivatives for more favorable results regarding hepatocyte cell injury. The authors found a combination of histidine and N-α-acetyl-L-histidine together with the addition of a membrane-permeable iron chelator to be feasible to ameliorate the original HTK solution. Further in vitro studies focused on the rate of cell survival, cell morphology and metabolic function of hepatocytes found especially the concentration of 0.5 mM deferoxamine + 20 µM LK 614 to improve prolonged cold storage (4 °C, 1 week) which offers a potentially extended time frame for cell transplantation without major loss of function [30].

Most in vivo studies utilizing animal models for liver transplantation could confirm the advantages of the HTK-N solution.

First, animal studies intended to analyze the systemic toxicity as well as hemodynamic and microcirculatory consequences of HTK-N [29] and revealed no adverse effects. The subsequent analysis of preservation and reperfusion injury showed and increasing gradient of morphological alterations with increasing storage period in both HTK and HTK-N solution. However, HTK-N led to an attenuated preservation injury after cold preservation for 24 h. In addition, the improvement of organ preservation was demonstrated by decreased apoptosis index and improved bile production (24 h cold storage and 60 min of reperfusion) in rat livers stored in HTK-N when compared with HTK.

In 2009, Wu et al., compared the effect of HTK-N with or without the addition of iron chelators to the traditional HTK solution in an isolated rat liver perfusion model [31]. The analysis of liver damage via LDH revealed a significant lower effect during reperfusion after storage of liver grafts in HTK-N with iron chelators when compared with HTK solution. Similarly, total bile secretion as an indicator for liver function was again significantly higher in the HTK-N with iron chelators group compared with traditional HTK. Histomorphological investigation (less hepatocyte vacuolization and edematous changes) and evaluation of microcirculation could confirm the improved preservation in HTK-N with iron chelators.

Further studies analyzing the components of HTK-N tested a chloride-poor and a chloride-containing variant of HTK-N in a rat liver transplantation model [48]. Liver transplantation was performed orthotopically according to the cuff technique described by Kamada and Calne [61]. The study was subdivided into two parts to analyze (a) the overall survival under different conditions (24 h, 12 h and 3 h of cold ischemia time) and (b) microcirculation and laboratory/histological parameters. The authors observed a strong tendency for prolonged survival after graft preservation with chloride-containing solution. Moreover, the postoperative intrahepatic microcirculation was significantly improved, and bile production significantly increased in the chloride containing HTK-N solution.

To evaluate livers from extended criteria donors, steatotic rat liver grafts were analyzed regarding protective effects of HTK-N [49]. Microvesicular steatosis was induced by a single dose of ethanol (8 g/kg BW). After storage of liver grafts at 4 °C for 8 h with HTK or HTK-N an arterialized orthotopic liver transplantation was performed with sample collection after 1, 8 and 24 h of reperfusion. Survival was compared after 1 week. This study showed serum liver enzymes (aspartate aminotransferase (AST), alanine aminotransferase (ALT), LDH) and liver injury (less necrotic areas and leukocyte infiltration) to be reduced. More importantly, animals of the HTK-N group had increased survival (87.5%) compared to the HTK group (12.5%) after one week of observation.

Another approach tested HTK-N for its suitability for aerobic preservation modalities in hypothermia for marginal donor livers from non-heart beating donors using either hypothermic machine perfusion (HMP) or the technique of venous systemic O_2_-persufflation (VSOP) compared to conventional static cold storage [47]. Rat livers were harvested 30 min after cardiac arrest. After 18 h of cold storage (4 °C) reperfusion was conducted in a circulating system for 120 min. VSOP with HTK-N minimized enzyme release (ALT, LDH), increased recovery of energy charge and led to improved functional recovery of the organ.

In a comparable set-up, HMP was performed for 18 h at 4 °C with the following preservation solutions oxygenated with 100% O2: HTK; HTK-N without additives or with the addition of 25 µM deferoxamine + 2.5 µM or 7.5 µM of the permeable iron chelator LK 614 [38]. HTK-N provided protective effects reflected by lower levels of post reperfusion ALT and LDH levels. Evaluation of different quantities of the iron chelator LK 614 resulted in improvement of metabolic activity measured by hepatic production of CO_2_ upon reperfusion when using HTK-N with the addition of 2.5 µM of LK 614.

#### 2.2.2. Kidney

In contrast to the evaluation in liver transplantation, studies assessing HTK-N in the residual abdominal organs (e.g., kidney) are rare.

A recently conducted study performed kidney transplantation with conventional SCS of kidney grafts and preservation in either UW, HTK or HTK-N solution for 30 h at 4 °C in a porcine model [52]. The animals were postoperatively monitored for 7 days with blood collection on day 3 and 7, and biopsies were taken directly after cold ischemia time, 30 min after reperfusion, and on the 7th post-transplant day. Despite the above-mentioned auspicious results revealed for liver preservation, no statistical significance was found regarding the parameters of graft function and histopathological changes for kidneys. The authors assumed that the advantages of HTK-N compared to HTK and UW as described in other organs might become measurable in clinical settings where additional factors such as old-for-old organ, extremely long ischmia time or marginal organs demand more specific requirements of a preservation solution.

In a different experimental setting, porcine kidneys were preserved with HMP [50]. The authors utilized an HTK-N variant supplemented with dextran that offered superior protection for lung transplantation in a previous study [46]. Kidneys were preserved for 20 h by pulsatile oxygenated HMP on a Lifeport kidney transporter (syst. pressure 30 mmHg, 30 cycles/min) with either HTK-N supplemented with 50 g/L dextran 40 or kidney perfusion solution 1 (KPS-1). In general, renal function was significantly improved when preserving kidneys with HTK-N plus dextran compared to KPS-1 demonstrated by higher clearance values of creatinine and higher metabolic activity of the kidneys (postischemic oxygen consumption). These preliminary results of isolated organs were then confirmed by a follow-up study in a porcine autotransplant model [51]. After procurement, grafts were preserved overnight (21 h) by pulsatile perfusion in the Lifeport Kidney Transporter System using either HTK-N supplemented with dextran 40 or KPS-1 as gold standard. Postoperatively animals were observed for 7 days. Oxygen free radical-mediated tissue injury and tubular cell injury was significantly higher in KPS-1 compared to HTK-N plus dextran. Measurement of endothelial activation markers and proinflammatory cytokines suggested animals receiving kidneys preserved in HTK-N plus dextran to exhibit less endothelial stress response. The authors suggested HTK-N plus dextran to potentially minimize transient early dysfunction after transplantation especially of interest upon more critical preservation times or for marginal donor organs

#### 2.2.3. Pancreas

Until now, solely one study was conducted evaluating the impact of HTK-N in the preservation of pancreas grafts [53]. Esmaeilzadeh et al. used a porcine model of heterotopic pancreas transplantation, in which during procurement grafts were flushed and subsequently stored for 10 h at 4 °C in UW, HTK or HTK-N solution. Following a duodenum preserving total native pancreatectomy in recipient animals, grafts were transplanted heterotopically and animals were monitored for 7 days. With this rather prolonged observation period the authors were able to not only evaluate direct effects of cold storage induced injury but also to monitor insulin and glucose levels as surrogate markers for pancreas function during recovery phase. However, no statistical differences were found for serum glucose and insulin levels as well as histopathological injury when comparing HTK-N to HTK or UW. While this study could not reveal a superiority of HTK-N in comparison to UW or HTK, it was shown that HTK-N can be used to preserve pancreas grafts for 10 h with equal results as with the currently used standard HTK and UW solutions. The authors further speculate whether a longer and thus more clinically relevant cold storage time would be needed to detect protective effects of HTK-N and such studies are expected.

#### 2.2.4. Small Bowel

Although isolated small bowel as well as multivisceral and modified multivisceral transplantation have evolved to an established therapeutic option for patients with intestinal failure, there remain major obstacles for this lifesaving operation [62]. Among the most critical issues is the susceptibility of intestinal grafts to ischemia [63]. As its ultimate long-term curative effects are not comparable to other organ transplantations, this delicate organ might profit from an amended preservation solution.

There are currently two animal studies testing the influence of HTK-N on intestinal grafts. Chen et al. compared HTK and HTK-N in an ex vivo model of warm oxygenated reperfusion of isolated intestinal rat grafts following 8 h of static cold storage [55]. This study is of special interest since it also compares different preservation routes by administrating both HTK and HTK-N as a vascular as well as additional luminal conservation solution. With regard to its endpoints, this study focuses on cold storage induced functional and structural alterations. An addition of luminal plus vascular HTK-N had positive effects on the integrity of intestinal mucosal lining in addition to an enhanced goblet cells vitality. However, the effects in this particular study were only minor.

Lautenschlager et al. used the more sophisticated model of heterotopic, syngeneic small-bowel transplantation in the rat after a cold storage period of 24 h [54]. Using intravital microscopy, this study was focused on alterations of the small-bowel mucosal microcirculation elicited by intestinal IRI. By comparing a total of nine different groups, the authors tested different concentrations of the iron chelators, deferoxamine and LK 614 as well as addition of glycine. Modified HTK (with glycine, alanine and iron chelators) was found to improve perfusion index, functional mucosal capillary density as well as red blood cell velocity in intestinal grafts when compared to HTK. This provides valuable inside into potential mechanism of HTK-N since it reveals changes in the mucosal microcirculation, an area severally effected by IRI. However, further studies adjusting cold storage and a postoperative observation period, as well as components of the solution might be recommended to ascertain potential beneficial effects of HTK-N on intestinal graft function and performance following preservation.

## 3. Clinical Studies

A prospective, randomized, single blind multicenter phase III study currently aims to demonstrate non-inferiority of graft preservation with HTK-N compared with HTK [64]. The study focuses on graft function and injury after transplantation of kidney, liver or combined kidney-pancreas. Primary measure outcomes are for kidney transplantation delayed graft function (dialysis requirement during first week after transplantation) and for liver area under the curve (AUC) and ALT after transplantation for 7 days. The estimated study completion date is May 2023.

Another study focuses on living donor kidney transplantation in a prospective, randomized single blind, monocenter phase II study comparing HTK-N with HTK [65]. The trial ended in April 2018. Primary outcome measures were renal function reflected through the glomerular filtration rate (GFR) calculated with CKD EPI (Chronic Kidney Disease Epidemiology Collaboration) GFR equation (3 months after transplantation). The results are not yet published.

A further prospective, randomized, single blind, multicenter phase III study analyzing HTK-N in the setting of heart transplantation was carried out from July 2016 until September 2019 [66]. Primary outcomes of the study were creatine kinase myocard type (CK-MB) peak value. Secondary outcome measures comprised of catecholamine requirement, patient survival and complications. Until now, results of the study are unpublished.

## 4. Outlook

Escalating organ shortage has further reinforced the continuing trend in modern transplant medicine to utilize marginal organs, including organs donated after circulatory death (DCD). While these organs clearly expand the donor pool, they impose an additional risk for the recipient. Therefore, use of HTK-N might be a reasonable strategy to ameliorate preservation injury in these vulnerable organs. HTK-N has been successfully used in two animal models of DCD liver [38] and kidney donation [67]. However, the available clinical data are limited, and DCD organs are excluded in the current multicenter clinical trial on HTK-N in kidney, liver and pancreas transplantation [64]. Parallel to the increased utilization of DCD organs is rise of machine perfusion in solid organ transplantation [5]. In this setting, HTK-N has been used in different settings including hypothermic machine perfusion of rodent liver grafts [38], pulsatile oxygenated hypothermic machine perfusion of porcine kidney grafts [50] as well as machine-controlled oxygenated rewarming of porcine kidney and liver grafts [67,68]. However, not all of these studies were designed for head to head comparison between HTK-N and other preservation solutions. Stegemann et al. compared HTK-N with HTK in a model of hypothermic machine perfusion of marginal liver grafts and could demonstrate significant improvements by HTK-N [38]. In line with this, HTK-N has been shown to be superior over a standard machine preservation solution (kidney perfusion solution1 (KPS1)) in kidney transplantation [51]. Thus, HTK-N is a promising tool for two of the main trends in transplant medicine, machine perfusion and utilization DCD organs.

## 5. Conclusions

In summary, HTK-N presents an auspicious alternative to current standard preservation solutions in solid organ transplantation. Due to several novel mechanistic insights into IRI, adjustments are needed to correspond to the increasing need for donor organs and for the acceptance of organs from extended criteria donors. Specific features of the individual organs of the abdominal and thoracic system might require unique adjustments of the solution to obtain the most favorable outcomes for organ preservation. Nevertheless, until now, no general recommendations for the usage of HTK-N and for the replacement of former gold standard solutions in solid organ transplantations can be made. Ongoing clinical trials, further studies in large animal models and consequently in humans will show its significance for the different organ systems relevant to transplantation.

## Figures and Tables

**Figure 1 ijms-21-06468-f001:**
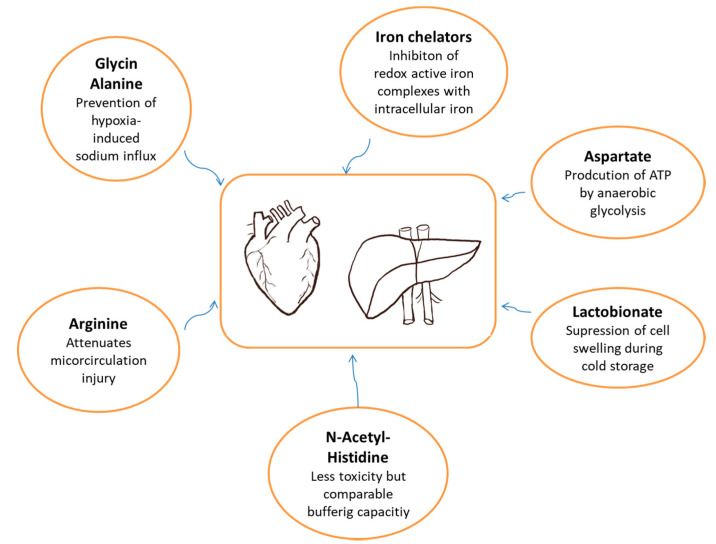
HTK-N and its site of actions on preserved organs.

**Table 1 ijms-21-06468-t001:** Composition of the different preservation solutions.

Components (mmol/L)	EC	CE	UW	HTK
**Electrolytes**				
Calcium		0.25		0.01
Chloride	15		20	50
Magnesium		13	5	4
Potassium	115	15	120	10
Sodium	10	100	30	15
**Osmotic Properties**				
Glucose	195			
HES			59 (g/L)	
Lactobionate		80	100	
Mannitol		60		30
Raffinose			30	
**Antioxidants**				
Allopurinol			1	
Glutathion		3	3	
Tryptophan				2
**Buffers**				
Histidine		30		198
KH_2_PO_4_	43		25	
K_2_HPO_4_	15			
NAHCO_3_	10			
**Additives**				
Adenosine			5	
Ketoglutarate				1
**Amino Acids**				
Glutamic acid		20		
pH	7.0	7.3	7.4	7.2
Osmolarity (mOsm/L)	355	320	330	310

EC = Euro-Collins; CE = Celsior; UW = University of Wisconsin; HTK = histidine-tryptophan-ketoglutarate.

**Table 2 ijms-21-06468-t002:** Composition of HTK and HTK-N.

Constituents (mmol/L)	HTK	HTK-N
Na+	15	16
K+	10	10
Mg++	4	8
Ca++	0.01	0.02
Cl-	50	30
Histidine	180	124
Histidine HCl	18	
N-acetylhistidine		57
Aspartate		5
Tryptophan	2	2
Oxoglutarate	1	2
L-arginine		3
Glycine		10
L-alanine		5
Saccharose		33
Manitol	30	
Deferoxamine		0.025
LK 614		0.0075

**Table 3 ijms-21-06468-t003:** Animal Studies Evaluating the Impact of HTK-N.

Organ	Study	Experimental Model	Evaluated Solutions	Results
Thoracic organs				
Heart				
	Koch et al., 2009 [40]	Heterotopic heart transplantation Lewis rats 1 h cold storage 1 h reperfusion	(1) HTK (2) HTK-1 = addition of l-arginine, N-acetyl-histidine, reduction of chloride	Left ventricular systolic pressure + minimum rate of pressure development higher in HTK-N
	Koch et al., 2010 [41]	Heterotopic heart transplantation Lewis rats 1 h cold storage 1 h reperfusion	(1) HTK (2) HTK-1 = addition of l-arginine, N-acetyl-histidine (3) HTK-2 = addition of deferoxamine, LK614	Left ventricular pressure + dP/dt minimal higher in HTK-1
	Loganathan et al., 2010 [42]	Heterotopic heart transplantation Lewis rats 1 h and 24 h reperfusion	HTK HTK-N	HTK-N: Better myocardial relaxation and coronary blood flow after 1 h reperfusion Better apoptosis index and energy charge potential after 1 h and 24 h of reperfusion
	Wu et al., 2011 [43]	Heterotopic heart transplantation C57BL/6J mice 24 h cold storage Reperfusion up to 2.5 months (survival analysis)	(1) N44: chloride poor (2) N45: chloride rich → With or without deferoxamine (100 µmol/L), LK614 (20 µmol/L) (3) N46: superior variant with or without iron chelators (100 µmol/L deferoxamine 10/20 µmol/L LK614)	N45 + iron chelators longer survival N46 and N46 100/10: Shortened re-beating time Less interstitial oedema Decreased enzyme release N46: reduced apoptotic index N46 100/10: Less inflammatory infiltration Better myocyte preservation Less oxidative stress
	Turk et al., 2012 [44]	Heterotopic heart transplantation C57BL/6J (H-2b) mice 15 h cold storage 60 days reperfusion	HTK N46 (HTK-N including deferoxamine and LK614	N46: better re-beating time, palpation score and cardiac fibrosis
	Schäfer et al., 2019 [45]	Isolated hearts Guinea pigs (1) 80 min ischemia 30 °C (2) 81 min ischemia 30 °C with intermittent perfusion (3 × 4 min) of cold perfusion solution (5 °C) (3) 360 min ischemia at 5 °C Reperfusion 45 min in vitro	(1) HTK (2) HTK-N (3) iHTKCa80 (HTK + 0.05 mmol/L Calcium) (4) iHTKCaN80 (HTK + 0.03 mmol/L Calcium)	5 °C iCa^++^ at t-in higher than before ischemia HTK: longest t-in, shortest t-ret, best VS-RR, higher LVDP 30 °C Intermittent ischemia HTK-N shortest t-ret, best VC-RR, highest LVDP
Lung				
	Pizanis et al., 2012 [46]	Single-lung transplantation German Landrace pigs 24 h cold storage 6 h reperfusion	(1) LPD (2) HTK-N (3) HTK-N without iron chelators (4) HTK-N plus dextran 40	HTK-N plus dextran 40: Reduction of mean pulmonary arterial pressure and pulmonary vascular resistance Increased oxygen capacity and reduced pCO_2_ levels Reduced oxidative stress Compared with LPD
Abdominal organs				
Liver				
	Bahde et al. 2008 [29]	Isolated livers Lewis rats (1) Cold storage 60 min, 12 h, 24 h, 48 h, 4 d, 7 d (2) 24 h or 72h cold storage and 60 min in vitro reperfusion	(1) HTK (2) Modified HTK (at that point without iron chelators)	Modified HTK: After 4 d and 7 d of cold storage attenuated preservation injury Decreased LDH release after 72 h cold storage and during reperfusion Increased bile production after 24h cold storage Lower apoptotic index after 24 h cold ischemia and 60 min reperfusion
	Wu et al. 2009 [31]	Isolated perfused livers Wistar rats 24 h cold storage 30 min/60 min in vitro reperfusion	(1) HTK (2) Modified HTK (3) Modified HTK plus 20 µm LK614	Modified HTK + LK614: Decreased LDH release during reperfusion Increased bile secretion Better preserved hepatic microcirculation
	Stegemann et al. 2009 [47]	Isolated livers from non-heart beating donors Wistar rats 18 h cold storage 120 min in vitro reperfusion	(1) HTK (static cold storage) (2) HTK-N (static cold storage) (3) HTK-N (HMP) (4) HTK-N (VSOP)	HTK-N Less free radical mediated lipid peroxidation Improved enzyme leakage upon reperfusion and histological integrity HTK-N +VSOP Minimized enzyme release and histological damage Increased bile production and energy charge
	Stegemann et al. 2010 [38]	Isolated livers from non-heart beating donors Wistar rats 18 h cold storage HMP 120 min in vitro reperfusion	(1) HTK (2) HTK-N (3) HTK-N + 25 µm deferoxamine + 2.5µm LK614 (4) HTK-N + 25 µm deferoxamine +7.5 µm LK614	HTK-N and its variants decreased of enzyme release (ALT, LDH) HTK-N + 25 µm deferoxamine +2.5 µm LK614 improved metabolic activity, reduced cleavage of caspase 9 and apoptotic index
	Fingas et al. 2011 [48]	Orthotopic liver transplantation Lewis and Wistar rats Cold storage 24 h, 12 h, and 3 h Reperfusion up to 28d	(1) HTK-N chloride-poor (0.004 mmol/L) (2) HTK-N chloride-containing (34.04 mmol/L)	HTK-N chloride-containing: Prolonged survival Ameliorated microcirculation Increased bile production
	Liu et al. 2012 [49]	Orthotopic liver transplantation Beforehand induction of microvesicular steatosis Sprague-Dawley rats Cold storage 8h Reperfusion up to 1 week	(1) HTK (2) HTK-N	HTK-N Prolonged survival Decreased liver enzymes, necrosis, leukocyte infiltration and MPO Less caspase-3 and iNOS expression
Kidney				
	Gallinat et al., 2013 [50]	Isolated kidneys German Landrace pigs 20 h cold storage HMP Reperfusion 120 min in vitro	(1) HTK-N plus 50 g/L dextran 40 (2) Kidney perfusion solution (KPS)	HTK-N plus 50 g/L dextran 40: Higher renal blood flow and urine production Increased metabolic activity Increasing clearance of creatinine
	Minor et al., 2015 [51]	Autologous kidney transplantation Landrace pigs 21 h cold storage HMP Reperfusion 1 week	(1) HTK-N plus 50 g/L dextran 40 (2) Kidney perfusion solution (KPS)	HTK-N plus 50 g/L dextran 40: Better microcirculatory tissue perfusion Less oxygen free radical-mediated tissue injury and tubular cell injury Decreased endothelial stress response Better clearance function during first 24 h after Tx
	Golriz et al., 2017 [52]	Allogene kidney transplantation Landrace pigs 30 h cold storage 7 days of reperfusion	(1) HTK (2) HTK-N (3) UW	no statistical differences between groups
Pancreas				
	Esmaeilzadeh et al., 2015 [53]	Allogene pancreas transplantation Landrace pigs 10 h cold storage 7 days reperfusion	(1) HTK (2) HTK-N (3) UW	Higher insulin levels in UW group
Small Bowel				
	Lautenschlaeger et al., 2018 [54]	Heterotopic small bowel transplantation Syngeneic Lewis rats 24 h cold storage 180 min reperfusion	(1) HTK (2) HTK + glycine (10 mM) (1) HTK + deferoxamine (1 mM) (2) HTK-N + deferoxamine (0.5 mM) (3) HTK-N + deferoxamine (0.2 mM) (4) HTK-N + LK614 (0.05 mM) (5) HTK-N + LK614 (0.02 mM) (6) HTK-N + deferoxamine (1.0 mM) + LK614 (0.05 mM) (7) HTK-N + deferoxamine (0.2 mM) + LK614 (0.02 mM)	HTK-N + iron chelators ameliorated microcirculatory parameters (perfusion index, functional capillary density of the mucosa, and red blood cell velocity)
	Chen et al., 2020 [55]	Isolated small bowel Lewis rats 8 h cold storage (1) Vascular flush with preservation solution (2) Vascular plus luminal flush with preservation solution 30 min reperfusion in vitro	(1) HTK (2) HTK-N	HTK-N vascular route decreased ultrastructural alterations HTK-N plus additional luminal route ameliorates integrity of intestinal mucosa, vitality of goblet cells and intestinal edema

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
