# Peer review of "HTK-N: Modified Histidine-Tryptophan-Ketoglutarate Solution—A Promising New Tool in Solid Organ Preservation"

_ijms, 2020, doi:10.3390/ijms21186468_

Round 1

Reviewer 1 Report

This is a very well written and focus review. This has provided an overview of the current solid-organ preservation using the novel HTK-N solution. Several minor points detailed below.

Line 42. …secondary injury phase during reperfusion (inflammatory response). The reperfusion injury is not only the inflammatory response. It constitutes an array of responses, at least in the heart, such as ROS production, calcium overload, quick correction of pH, mitochondrial dysfunction and inflammatory responses (1, 2).

For heart transplant, what is the current status of organ preservation? Will HTK-N prolong the time, and by how much? What is it compared with a constantly perfused donor heart (up to 12 hours after harvest)? (3).

It will be helpful if the authors can provide a figure to depict which component of HTK-N affect which ischemia-reperfusion injury process/processes.

1.            Hausenloy DJ, Yellon DM. Ischaemic conditioning and reperfusion injury. Nat Rev Cardiol. 2016;13(4):193-209. Epub 2016/02/05. doi: 10.1038/nrcardio.2016.5. PubMed PMID: 26843289.

2.            Kloner RA, Brown DA, Csete M, Dai W, Downey JM, Gottlieb RA, Hale SL, Shi J. New and revisited approaches to preserving the reperfused myocardium. Nat Rev Cardiol. 2017;14(11):679-93. Epub 2017/07/28. doi: 10.1038/nrcardio.2017.102. PubMed PMID: 28748958; PMCID: PMC5991096.

3.            Monteagudo Vela M, Garcia Saez D, Simon AR. Current approaches in retrieval and heart preservation. Ann Cardiothorac Surg. 2018;7(1):67-74. Epub 2018/03/02. doi: 10.21037/acs.2018.01.06. PubMed PMID: 29492384; PMCID: PMC5827116.

Reviewer 2 Report

Dear Authors,

thank you for allowing me to revise your original manuscript entitled: HTK-N: Modified Histidine-Tryptophan-Ketoglutarate Solution A Promising New Tool in Organ Preservation.

The study makes a broad overview of a recently revolutionized topic in transplantation that represented one of the first challenges of transplant medicine and research: how to preserve cellular integrity and organ function during the ischemic time. The field has been based for over 50 years on static cold storage and HTK has been one of the most appreciate solutions whose composition has been studied to preserve the metabolic integrity of the cells. The entry of the ex-situ perfusion as an alternative philosophy to preserve organ function has speeded up the research on a new composition of HTK called HTK-N.

The manuscript describes the actual limitations relative to IRI onset during transplantation and how this new receipt may represent the answer to an unmet clinical need derived by the exigence to expand donor pool toward marginal donors. The topic is fairly described from the viewpoint of the transplant of all the more commonly transplanted organs. The rationale of the modification of the new composition is clearly described and offers a wide landscape of how the solution has been projected to interfere with metabolism during the ischemic phase. 

  • A figure with the metabolic steps and the site of action of the new additives could offer to the reader an incisive overview and could improve the scientific sound of this publication.

The part on the animal studies in thoracic transplantation and in abdominal transplantation appears full of data and of messages and quite complete. 

  • I would suggest the authors add a table containing all animals' studies (separating all the single organs) and clinical studies and schematically fixing the design, the objectives, the results, and the eventual drawbacks.

Before conclusions, I suggest adding a paragraph on the possible utility or even finality of such a solution in light of the evolution of the Transplant surgery toward the usage of DCD donation and machine perfusion. 

Best Regards

Round 2

Reviewer 2 Report

Dear Authors,

compliments for the improvements performed to the manuscript according to reviewers' suggestions.

Best Regards